# Health Service Impacts and Risk Factors for Severe Trauma in Mountain Biking: A Narrative Review

**DOI:** 10.3390/healthcare11243196

**Published:** 2023-12-18

**Authors:** Gillian Course, James E. Sharman, Viet Tran

**Affiliations:** 1School of Medicine, College of Health and Medicine, University of Tasmania, Hobart, TAS 7000, Australia; v.tran@utas.edu.au; 2Department of Emergency Medicine, Royal Hobart Hospital, Hobart, TAS 7000, Australia; 3Menzies Institute for Medical Research, College of Health and Medicine, University of Tasmania, Hobart, TAS 7000, Australia; james.sharman@utas.edu.au

**Keywords:** accident injury, trauma, severity index, emergency treatment, accident prevention

## Abstract

Mountain biking is growing in participation but carries risk for severe injury and burden on health systems. Little is known about the impact of these injuries on emergency medical services, definitive healthcare, and factors contributing to accidents. This review aimed to determine the health service impacts of severe mountain bike trauma and risk factors, with a view to understanding critical gaps and needs. A systematic online search was conducted using the databases *PubMed* and *MEDLINE complete* and grey literature relating to mountain bike injury since the databases’ inception to July 2023. The results show that although mountain biking has relatively high injury rates that are increasing, the impacts on health services were rarely documented, with some evidence indicating that even small increases in injuries from race events can overwhelm local health services. Severe injuries were more common in downhill disciplines. However, the definitions of what constitutes severe injury were variable. Severe injuries were more common in downhill disciplines, influenced by the rider skill level, demographics, participation in competitive events, trail design, environmental factors, and healthcare availability. Further research in these areas is needed, along with the more consistent reporting of injury severity.

## 1. Introduction

Cycling has seen increased participation in recent years, accompanied by a rise in related injuries. In Australia, collective data on cycling participation has shown a notable increase from 2017 to 2021 [1], suggesting that the COVID-19 pandemic, among other factors, corresponded with a significant surge in cycling participation and other individual sports. Data on injury-related hospitalisations in Australia for the same time period saw injuries caused by cycling increase by 57% [1], associating cycling with the highest number of recorded hospitalisations of any sport. No subcategories or injury severity were identified in the Australian report [1], but of particular concern was the higher incidence of intracranial and internal/vessel of trunk injuries in cycling compared to other sports. While these injuries are presumably severe, the data do not provide specific details of the cause, severity, or outcomes. This narrative review specifically focuses on injuries in mountain biking (MTB), a subcategory of cycling, to shed light on this aspect of the sport.

MTB refers to off-road cycling, often on steep and rough terrain [2]. Whilst it is challenging to quantify participation due to inconsistent data collection methods, the increasing global sales of mountain bikes suggests that it is now one of the world’s fastest-growing recreational sports [3] and has become important for local economies and tourism [3]. However, MTB is an inherently risky sport, and the growing participation is likely to lead to higher rates of injury, potentially imposing a significant healthcare burden and public health concern. Despite the growth of the sport and the high risk of injury, there is a lack of research on the incidence and impact of MTB injuries on health services.

After cross-country MTB debuted in the 1996 Atlanta Olympics, the demand for mountain bikes escalated, rapidly increasing their market share [4]. The subsequent development of off-road bike technology [5] and trail design led to the development of subdisciplines within the sport (Table 1). The advent of electric/pedal assist mountain bikes (eBikes) has further expanded the appeal of MTB, enabling people of different ages and fitness levels to cover greater distances on steep uphill terrain with minimal effort. The popularity of eBikes is evident in the doubling of sales in 2020 [6].

MTB has evolved into a lucrative industry [7] with the establishment of more than 250 international purpose-built trail developments known as MTB ‘parks’ [8]. In Australia, there was over $100 million in government funding approved in 2020 for MTB park development [9]. These parks often offer trails of varying difficulties, and amenities such as bike rental, first aid, and shuttles for easy access to steeper downhill trails. Safety is not regulated in MTB parks but may be enhanced through trail design, warning signs at high-risk features, and community education. Many ski-resorts have found that developing MTB trails for off-season use is a profitable source of revenue in the summer months, attracting a high density of riders [10]. The increasing trend of MTB parks, especially those focused on gravity riding, may introduce new risks to MTB [5] and place additional demands on healthcare resources, particularly those in the local vicinity [11]. However, there is a paucity of evidence regarding the patterns of risk related to MTB injuries and their influence on hospital visits and admissions. 

For riders, making informed safety decisions such as riding within one’s level of ability, selecting a suitable trail, and opting to wear protective equipment may help to mitigate the risk of injury [12,13]. The use of protective equipment varies across different MTB disciplines (see Table 1) but personal safety in MTB is primarily the responsibility of the rider. Helmet use is mandatory in only four countries globally, and MTB trail classifications varies from country to country. The International Trail Marking System, which is universally used in ski fields, has been adapted at some MTB parks to classify trails based on difficulty [14], relying on riders’ self-assessment of fitness and capabilities (Table 2). The risk of injury in MTB may be influenced by rider experience, riding style, and trail surface, with varying levels of risk among experienced, recreational, and competitive riders [13,15,16,17]. In comparison to road cycling, MTB fatalities are uncommon [18,19], primarily because MTB trails are typically free from motor vehicle traffic. In an Australian report based on national hospital morbidity data from 1999 to 2016, off-road crashes accounted for only a small percentage [4.3%] of pedal cyclists deaths per year [19]; however, MTB mortality related to head injury is a significant concern [20,21].

The more ‘extreme’ MTB subdisciplines downhill and freeride are frequently referred to as ‘extreme’ sports [18,19] and carry the highest risk of severe injury [16], yet the understanding of these risks is limited. Increases in severe injuries are likely to have a significant impact on health services. This review aims to determine the health service impacts of severe MTB injuries and associated risk factors for severe MTB injury, with a view to understanding the critical gaps and needs. This information can contribute to the development of best practice in healthcare response and resource allocation and enhance the safety profile of MTB to assist in mitigating the risk of severe injury and alleviate the healthcare burden.

## 2. Methods

A systematic literature search was conducted using databases *PubMed* and *MEDLINE complete* and grey literature relating to MTB injuries since the databases’ inception to July 2023. Only full-text articles written or translated into English were used. Studies on cycling injury where MTB was not specifically noted as a key term were excluded from this review.

Papers were assessed by a single researcher using a Boolean search of electronic peer-reviewed journal literature using the following search terms: Mountain bik*, injur*, trauma, health service, hospital, trauma centre. A search of grey literature on Google using keywords (MTB, injury, emergency, healthcare) was conducted to identify government and industry studies, reports, and websites of relevant organisations.

Titles, abstracts, and full text were assessed for relevance to the study aim. Reference lists of full text documents were also searched for relevant articles to include. A total of 56 sources including original articles, review articles and online sources were analysed.

Trauma scoring systems are used to indicate injury severity, the risk of mortality and assist in predicting patient management [22]. However, as no single international trauma scoring system exists without limitations, for the purposes of this study, papers on MTB injury were included irrespective of the measure of injury severity used.

## 3. Results

Including review articles, there were sixteen papers contributing to the body of literature on severe MTB injury (Table 3). Among these, four papers specifically identified healthcare concerns in their discussion [21,23,24,25], and one paper concluded that MTB injuries have implications for healthcare resource allocation [20]. No articles presented specific information regarding the impacts of MTB injuries on healthcare systems. Several studies conducted in Canada, the US, and Europe utilised retrospective medical chart data and trauma registry information to profile MTB injuries in areas with a high density of off-road trails [4,21,26,27,28,29]. Additionally, some research examined injuries specific to MTB parks using data from individual local health clinics [13,23]. Diverse definitions of severe injury and variations in research methods [5,13,17,23,27,30,31] prevented us from being able to draw conclusive associations between risk factors and severe MTB injury.

## 4. Discussion

### 4.1. Epidemiology of MTB Injuries

The epidemiology of MTB injuries provides insight into the nature of severe injuries and identifies trends and potential risk factors. This information is useful for future improvements in the overall safety of mountain bikers and the response of healthcare services. The most common injuries in MTB were overuse and minor injuries, followed by fractures of the clavicle and upper limbs [21,23,31]. Although severe traumatic injuries were relatively infrequent, they carried a high risk of morbidity and mortality, primarily associated with head, neck, and chest trauma [23,28,29,36]. Fatalities associated with MTB were rare, ranging from an estimated mortality rate of approximately 2 out of 1 million riders [21], to 3.5 per million riders [16]. Three studies focused on serious injuries [13,21,28], and found that the severe injury rates in MTB were already at the higher end compared to other outdoor sports [8]. For example, the incidence of cervical spine fractures was second highest in MTB (13 per 1,000,000 person-years) behind surfing (38 per 1,000,000 person-years), significantly more than in other outdoor pursuits such as skiing and snowboarding (5 and 7 per 1,000,000 person-years, respectively) [38].

Demographic factors, including sex and age, were associated with the risk, prevalence, and patterns of injury severity in MTB [12,39]. However, age-specific injury rates, especially in the more extreme subdisciplines of MTB, were underreported [2]. Women constituted a smaller proportion of the MTB population [27], but had higher overall injury rates to men. An 8-year study of competitive MTB injuries recorded the injury rates in women, which were almost double those in men [40]. The injury severity and type varied in different race disciplines, with women more likely to suffer fractures compared to men [12,27,40] and also to experience a higher incidence of falls over the handlebars [4,8,41]. This mechanism of injury was attributed to a lighter body weight, but further research is necessary to understand how factors like musculoskeletal differences, body weight, and bone density might influence sex-related MTB injury patterns [12]. Similar considerations may apply to younger riders, whose immature musculoskeletal development could affect injury severity [42]. Among males, those aged 30–39 years were the most commonly injured [12,27], but the 21–30 year age group had a higher number of trauma centre admissions [21], suggesting a higher severity index in this age group. In off-road bicycle crashes, generally, life-threatening injuries were more common in the over 45 age groups [19]. It is unknown if variations in injury rates could be associated with increased risk-taking behaviour or more aggressive riding in certain age groups. Injury rates vary across the age spectrum, and as participation in MTB increases, monitoring the injury burden in different demographics, subdisciplines, and MTB racing will be useful in understanding the associated risks [43].

Gaining insight into morbidity and mortality associated with MTB can help in developing injury prevention strategies [21], especially where deliberate risk-taking is at times integral to the sport. This literature search found one example where concern over severe MTB injury directly resulted in the implementation of a successful public heath campaign [21]. Following a Canadian trauma registry review [21] identifying high serious MTB injury rates, physicians, stakeholders, and MTB community representatives collaborated to produce an injury prevention program and media campaigns promoting safe riding practices. The local MTB community took proactive measures to enhance safety by implementing trail difficulty ratings and rider skills education; however, no specific information regarding the impact of these measures on the injury rates was reported.

### 4.2. Health Service Impacts

Severe injuries have a significant impact on healthcare services, necessitating detailed information on common injuries to allocate staff, resources, provide education, and prepare for competitive events. The goal of the healthcare management of severe trauma is to respond appropriately to ensure optimum and timely care [38]. This may involve various aspects such as evacuation, acute treatment, rehabilitation, future medical care, and social support [44]. While there is potential for MTB injuries to significantly impact emergency departments and hospital admissions [18], the cost of injury and consideration of how these injuries affect retrieval and local health services is not evidenced in the literature.

#### 4.2.1. Pre-Hospital

As MTB trails tend to be in sparsely populated areas, the evacuation process of MTB trauma presents logistical complexities comparable to other mountain traumas [45]. These challenges potentially result in significant costs. It is essential for medical teams to possess the skills and equipment to manage a wide range of injuries, including severe cases [5], particularly during MTB races, where there is often a corresponding increase in demand for health services [41]. To ensure timely and critical care in the event of a life-threatening injury, clear coordination pathways for patient retrieval to major trauma centres should be established by competition organisers [5]. While studies have explored prehospital characteristics, retrieval times, and their association with clinical outcomes in trauma experienced in mountain regions [46], there is limited research in the context of MTB injuries. Only three studies included in this review, all conducted in Canada [23,28,34], reported the mode of transport to hospital from the injury location, and only two of these reported the time of MTB injury to the time of arrival to a trauma centre [28,34]. Details on the time delays from the accident to first aid or difficult extrication were rarely available [34].

There are limitations in the collection of comprehensive healthcare data on MTB injuries. Internationally, studies that have examined health service demands associated with MTB injuries have primarily focused on presentations at a single health facility near MTB parks [11,23]. However, this focus may not capture data on seriously injured riders who are transferred to trauma centres but still contribute to demand on local emergency resources [23]. First aid clinics at MTB parks or events may only handle minor injuries, and riders with serious injuries may seek medical attention elsewhere [13] or require transfer to another acute care hospital [19]. Furthermore, for various commercial reasons, MTB parks may be reluctant to release injury and usage data [8,23].

#### 4.2.2. Acute Care

In retrospective hospital chart reviews, diverse terminology used in the initial documentation of patients with cycling injuries makes it difficult to identify MTB-related cases. This inconsistency presented a challenge in the interpretation of the data for this review, as the true numbers of patients with MTB injuries may not have been captured or not reported separately to other cycling injuries. A 5-year Australian study [18] conducted on emergency department presentations to two major city hospitals found that only 6.6% of the cases had the nursing triage assessment specify whether the patient presentation was from a road cycling or MTB accident [18]. By omitting important characteristics such as the speed, type of riding, and location in the triage reporting [18], the incidence of MTB injuries may be underestimated. Despite the likely significant cost and impact of severe injury on healthcare services, a true picture of the injury severity in MTB was difficult to obtain due to the heterogeneity of the definitions used to describe injury severity. Some studies used the Injury Severity Score (ISS) [21,28,32,33,34], while others reported the severity based on the hospital admission [4], retrieval to a trauma centre [23], or duration of absence from riding or racing [24] as impact outcome variables.

#### 4.2.3. Long Term Care

The health service cost of medical care is rarely addressed in studies on MTB injuries. Previous research conducted in 2009 at Whistler Bike Park in Canada [23], which examined the MTB injuries at a single health clinic, provided an estimate of the maximum patient medical cost at CAD2670, with a median cost of CAD970. Severe injuries, particularly traumatic brain and spinal cord injuries [38], incur a high social and economic cost that was not captured in the Whistler study, as complex trauma care occurred elsewhere. In developed countries, the median health cost of trauma care per patient ranged from USD11, 819 to USD33, 701, with the cost influenced by the severity of injury, mode of transport to hospital, age, medical treatment, and polytrauma [47]. The cost of treatment may extend beyond financial considerations, as serious injuries can impose significant personal costs on individuals, affecting many aspects of their lives [35,48]. To date, no research has explored the long-term healthcare costs associated with MTB injuries.

### 4.3. Risk Factors

Various factors were identified as potential risk factors for MTB injuries, including poor rider skills, high speed, challenging terrain, equipment failure, weather conditions affecting trail quality, and younger or older ages [12,13,19,24,25,27,28,30,31,33,37,41,42,48]. Table 4 provides a summary of these risk factors together with the principal findings of this review. The causes of severe MTB injuries were infrequently recorded, with details on the contributing factors of crashes only available in about 40% of the studies that met the search criteria of this review. Gaining more insight into the prevalent risk factors for severe injuries could inform preventative strategies and improve the risk assessment and timely management of MTB injuries.

#### 4.3.1. Rider Skills and Safety on the Trails

Downhill MTB was consistently identified as posing the highest risk for injuries [12,16,49]. The most prevalent mechanism of MTB injury was a forward fall over the handlebars, which frequently occurs during downhill riding [12,50]. This type of fall has also been identified as the main contributor in severe injuries such as spinal cord injury and spinal fractures [34]. A retrospective study conducted at three major trauma centres in Canada [21] found the injury rates almost tripled over a 10-year period, with the number of severe injury cases [AIS ≥ 3] rising from fewer than 10 in 1992 to approximately 40 in 2002. While various factors could have contributed to the observed increase, including higher participation rates, the study specifically highlighted the expansion of downhill disciplines as a probable causative factor [21]. This finding was corroborated by a study conducted on the injury presentations at Whistler MTB Park in Canada, renowned for its downhill and freeride trails [23]. In a single summer season, 898 mountain bikers presented for medical attention, with a collective total of 1759 injuries [23].

Riding at an appropriate skill level was commonly cited as a way for mountain bikers to reduce exposure to injury risks [12,21,27,49]. A European study that surveyed downhill mountain bikers over one summer season found the majority of injuries were attributed to rider error (73%) and poor trail conditions (31%) [17]. Signage with a clear indication of the trail’s difficulty rating can assist riders in selecting trails within an appropriate skill level. Additionally, trail ratings could guide consistent construction practices to ensure trails accurately reflect the assigned difficulty classification. For example, trails designed to reduce speed into technical sections may help to mitigate the risk of severe injury [13] and contribute to a safer riding experience. Currently, many different MTB trail difficulty ratings are in use and there is no standardised safety standard to determine a minimum level of safety specifications for trail design and construction. Notably, formal trails and MTB parks typically adhere to MTB industry safety guidelines and use trail classifications; however, informal trails, built by non-professional trail builders such as local riders, may pose higher risks. No studies have directly compared the risk for riders between formal and informal trails.

Personal protective equipment was cited as being beneficial in preventing minor MTB injuries [27,49]; however, serious injuries can still occur, emphasising the need for rider caution and safety measures [37,39]. The availability of protective equipment, particularly helmets, is widespread in MTB [21], although not all studies have been able to report on usage associated with injury due to the inconsistency of documentation in hospital admission notes [23,28]. A 2011 Scottish study found that 31% of injured riders were wearing some form of protective equipment, with helmet use as high as 99% [27]. This high prevalence was concordant with other studies including helmet usage data [21,31,33,39]. Other protective items, such as knee pads, neck braces, and gloves were documented in several studies [21,24,27,29,30,34]. Compared to other adventure sports with high rates of traumatic injury, MTB has relatively low rates of head injuries, which may be attributed to mandatory full-face helmet use [4] in competitive events and some MTB parks [29].

#### 4.3.2. Risk in Racing Versus Recreational Riding

The impact of competitive MTB on the incidence of injury was unclear. Injury rates differed between disciplines, with downhill and enduro races seeing higher injury rates than cross-country MTB [31,51]. When comparing elite mountain bike riders to amateur riders during the Swiss Epic MTB race, elites had more exposure time due to additional training hours but did not have significantly more injuries than amateurs [37]. This finding concurred with other studies where no significant difference in injury prevalence between elite and amateur riders during race events was identified [30,37,52]. A study of elite enduro MTB competitors found a significantly higher incidence of injury during races versus practice sessions (38.3 injuries/1000 race hours vs. 3.6 injuries/1000 practice hours) [24]. This suggests that the risk of injury may be associated with competitive MTB, particularly in disciplines where the goal is to achieve high speeds over technical terrain. Downhill competitive MTB can reach speeds of up to 70 km/h, leaving little margin for rider error, especially in unpredictable trail conditions [20].

Existing larger-scale studies based on retrospective hospital data [4,21] have exclusively involved amateur riders; thus, it was not possible to make a comparison with competitive riders. The questionnaire studies from MTB events generally focused on self-reported injuries that may not require medical attention, excluding severe injuries and fatalities [30]. Knowledge of injury patterns in competitive events would be useful for local healthcare services to plan for potential increases in emergency presentations [25,35,41]. Indeed, there was a 28% surge in emergency department presentations during a single international MTB event in regional Scotland [11].

In one of the earliest studies on MTB injuries, data were collected from 3624 competitors participating in various disciplines during a 5-day race event at an MTB park in California, USA [31]. The study found that the overall risk of injury was similar across different MTB disciplines, but the mechanisms of injury varied among the race events. Downhill riders identified loss of control when turning as a significant mechanism, while cross-country riders frequently collided with other riders. Interestingly, over 80% of all the injuries occurred during downhill sections of the race, irrespective of the riding style [31]. These findings suggest that monitoring the course design and its association with certain types of injuries could be useful in supporting competitive event organisers to improve the course safety and provide adequate medical support in high-risk locations.

#### 4.3.3. Mountain Bike Parks

MTB parks were identified as a common source of traumatic injuries among recreational riders, with injury rates reaching up to 15 injuries per 1000 riders [27]. Two studies [13,23] examined the injuries occurring in the non-competitive MTB park setting, providing additional evidence that MTB parks are a significant location for serious injury, particularly among adolescents aged 10–19 years [13]. This age group was reported as being more vulnerable to head and traumatic brain injuries [2,4], possibly due to physiological immaturity increasing the risk of long-term neurological disability [42]. This risk was particularly significant in the subdisciplines of downhill and freeride, involving steep, fast terrain with jumps and gaps [5,23,36], features common to MTB parks. Shuttle-accessed gravity-focused MTB parks have improved trail accessibility for many riders, but this also potentially increases the risk of injury because riders can access more challenging trails and can complete more descents in a day [23].

The risk of serious injury in downhill MTB is high [12,17,52], but the relative injury rate and trauma severity differs between disciplines [8]. The development of trails, MTB parks, and bike technology [including eBikes] may impact risk factors contributing to injury and severity, but more research is needed to understand their effect on injury statistics. Despite a steady annual increase in eBike sales over the past decade [53], no studies have specifically explored the associated risks of injury in these riders. With the increasing number of older riders [19], research on the age-related risk of severe injuries [4] is needed, also taking into account how underlying chronic conditions may complicate the injury severity [8].

## 5. Future Directions

MTB is a high-risk sport compared to many other recreational activities. Ongoing data surveillance is integral to enhancing the safety profile of MTB, and to meet the growing and evolving participation in the sport. As MTB gains popularity across a wider range of demographics, there is a need for accurate and current injury data so that riders can make informed decisions about the risks of participation in the sport, and targeted injury prevention strategies can be implemented. Identifying current trends in MTB injuries requires a more consistent approach to hospital admission documentation [18] and the establishment of a reproducible definition for severe injury [37]. Evaluation and learning from the safety models within MTB parks can be applied to new MTB trail development. Research focusing on the various subdisciplines within MTB, particularly enduro, freeride, and downhill, where risk for injury appears highest, is important in identifying distinct risk factors for serious injury and developing appropriate healthcare management strategies. Understanding the causes of crashes will likely require a combination of quantitative and qualitative approaches [36], along with follow-up studies of patients post-injury. Positive steps in the implementation of effective risk management strategies in MTB trauma will require collaboration with community stakeholders, MTB parks, and healthcare services to ensure adequate resources are available to meet the growing demand [23].

## 6. Conclusions

Numerous studies have explored the incidence, characteristics, and demographics of MTB injuries, and noted their potential significant impact on healthcare. However, there is a lack of research investigating the specific healthcare impacts and whether increased participation in different subdisciplines exposes more riders to severe injury risks. Severe injuries can place strain on local healthcare services, where the coordination of care and timely treatment is needed to improve patient outcomes. Further studies are necessary to understand healthcare impacts, identify risk factors contributing to the severity of MTB injuries, and ensure effective risk minimisation, optimal treatment, and the safe development of this evolving sport. Figure 1 summarises the future directions and conclusions of this review.

## Figures and Tables

**Figure 1 healthcare-11-03196-f001:**
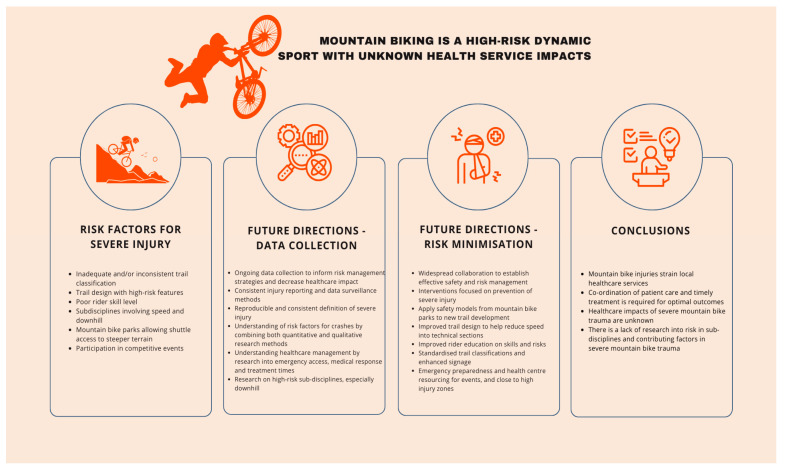
Summary of risk factors, future directions, and conclusions.

**Table 1 healthcare-11-03196-t001:** Defining characteristics of the subdisciplines of mountain biking.

Subdiscipline	Definition	Bike Features	Typical Trail Features	Bike Park Trail Grade	Typical Safety Equipment
Cross-country [XC]	Most common form of riding for beginners, at a competitive level combines fitness and bike handling skills	Any sort of mountain bike, often front suspension only	Single track, fire roads, gravel roads, sections of flat, down, and uphill, easier technical features	Green, blue	Open-face helmet, gloves
Enduro/All-mountain	Riding including climbs and descents, with more downhill focus	Slacker bike geometry, dual suspension, powerful brakes, bikes suited to ups and downs, heavy duty components	Challenging technical terrain on single track	Blue, black	Open- or full-face helmet, gloves, knee pads with other body armour optional
Downhill	High speed, gravity focused, usually all downhill with riders being shuttled to a highest point	Dual suspension, heavy duty components to cope with rough terrain, promote rider position for downhill only	Fast continuous steep downhill, often technical terrain with difficult obstacles, may include jumps, drops, and gaps that need to be cleared, common to mountain bike parks	Black, double black	Full-face helmet, knee pads, elbow pad, eye protection, body armour, neck brace, gloves
Freeride	Technical riding on natural terrain, made famous by ‘Redbull Rampage’	Strong, small, and light so manoeuvrable whilst airborne	Freeride is not on trails, often in rocky landscapes, riding off cliffs/slopestyle on manmade jumps, gaps, and ramps	N/A	Full-face helmet
Dirt jumping/slopestyle	Riding over dirt jumps and becoming airborne, doing tricks on man-made jumps and features, at a competitive level judged on style, elevation, originality	Small, light mountain bike or specially designed ‘dirt jumper’, may have front suspension	Jumps made of dirt, often part of a mountain bike parks, often no use of single track	N/A	Open-face or full-face helmet

**Table 2 healthcare-11-03196-t002:** Mountain bike trail difficulty rating according to International Mountain Bike Association (IMBA).

Trail Marking Colour	Difficulty Rating
White	Very Easy
Green	Easy
Green/Blue	Easy with intermediate sections
Blue	Intermediate
Blue/Black	Intermediate with difficult sections
Black	Difficult
Double black	Extreme

Source: International Mountain Bike Association standards.

**Table 3 healthcare-11-03196-t003:** Summary of literature on mountain bike injury rates, types, severity, and health care findings.

Author-Year	Study Years	Country/Location of Study/Focus	Study Design	Population of Injured Mountain Bike Riders Studied	Riding Discipline/Elite or Amateur	Types of Injuries Studied	M:F % (Mean Age)	Incidence of Mountain Bike Injury	Injury Severity Score Recorded (ISS)	Study Findings Relevant to Health Care Impacts
Aitken, Biant and Court-Brown 2010 [27]	2007–2008	Glentress, Scotland	Mountain bike-related emergency dept. presentations to 5 local medical facilities prospectively identified	202	All	All	83:17 (31.5)	1.54 per 1000 biker exposures	Not discussed as a separate variable, injuries classified by body region	Not discussed
Ashwell, McKay et al., 2012 [23]	2009	British Columbia, Canada	Injured mountain bike park riders admitted to Whistler Health Clinic in one summer season	898	All	All	86:14 (26)	NR	Severity based on the need for patient to be transferred to Level 1 trauma centre	Healthcare planning for peak injury times, improve rider focused injury prevention, safer trail design, longitudinally study cost of injury to assess what level of injury in mountain biking is acceptable
Beck, Stevenson et al., 2016 [32]	2013	Victoria, Australia	Injured cyclists admitted to two major trauma services using State Trauma Registry	28	Data includes all cyclists road and off-road, not a study of mountain biking exclusively	All	81:19 (35)	NR	ISS recorded as inclusion in State Trauma Registry	Considers the social cost of injury by correlating return to work time. Conclusions aimed at all cycling injuries with few mountain bikers included in this study
Chow and Kronisch 2002 [33]	1994–1998	California US	Interviews with injured riders at 7 national off-road racing events	97	Events included downhill, XC, trials, dual slalom, 4 crossamateurs	All	74:26 (28.3)	NR	ISS recorded for all injured riders, focus of this study was more severe injuries	Focus on mechanism of injury rather than injury rates, not focused on healthcare perspective
Dodwell, Kwon et al., 2010 [34]	1995–2007	British Columbia, Canada	Chart review of patient with spinal injuries related to mountain biking at Level 1 trauma centre, provincial spinal referral centre	107 all spinal cord and spinal fractures secondary to mountain bike riding	Off-road MTB, all disciplines, amateurs + 2 elites	Spinal injuries	95:5 (32.7)	Mountain biking related to 3.8% of spinal trauma admissions over 13 years	ISS noted	
Gaulrapp, Weber et al., 2001 [30]	NR	Europe	Questionnaires to subscribers of an international mountain bike magazine in Germany, Austria, and Switzerland	3474	Not specified	All	98:02 (25)	1.1 injuries per 1000 riding hours	Severity graded by clinical findings and time off from riding, 10% classed as severe, requiring more than 3 weeks recovery outpatient treatment or hospitalisation	
Jeys, Cribb et al., 2001 [35]	NR	Shropshire, UK	Prospective data from 1 year of off-road mountain bike presentations to an orthopaedic trauma unit of major hospital	84	Not specified	All	71:29 (22.5)	1.6 injuries per patient	6 patients with severe injury described	Impact on rural hospitals noted that did not pre-exist the recent increase in participation, need to understand mechanism of injury
Kim, Jangra et al., 2006 [21]	1992–2002	British Columbia, Canada,	Retrospective chart reviews of mountain bike-related admissions to 3 trauma centres in the Greater Vancouver area serving major mountain bike parks and trails	399	Not specified amateurs	All	NR	Mean of 0.9% of annual trauma admissions were mountain bike-related	Focus of study was serious injury or death, requiring trauma centre admission for 3+ days and inclusion on BC Trauma Registry	Injury prevention programs developed based on this study—stakeholder and mountain bike community engagement to develop a media campaign to promote safe riding habits
Kotlyar 2016 [29]	2012–2015	Southwest Colorado, US,	Retrospective chart review of all injuredcyclists to a resort medical centre in a major mountain bike destination	304	All off-road trail riding, disciplines not specified nor trail vs. bike park, amateurs	All	70:30 (NR)	67% of cycling injuries were mountain bike-related	NR	
Kshirsagar, Xiao et al., 2021 [36]	2009–2018	US	Retrospective analysis of mountain bike-related head and neck injuries via US Nationwide via National Electronic Injury Surveillance System (NEISS)	486	All	Head and neck	81.29 (35)	NR	Severity based on need for hospital admission	Not discussed
Nelson and McKenzie 2011 [4]	1994–2007	US	Retrospective analysis of mountain bike-related injuries treated in emergency departments from NEISS data	4624	All amateurs	All	81:19 (36.2)	6.2 per 100,000 population	Severity determined by need for hospitalisation and body region of injury	Not noted but this is a very large-scale study of burden of mountain bike injury
Palmer, Florida-James et al., 2020 [24]	2017–2018	International	Rider injuries recorded during two seasons of the Enduro World Series (EWS) events	179	Enduro elites	All	90:10	38.3 injuries per 1000 race hours, 3.6 per 1000 practice hours	Severity of injury based on time taken to return to riding in days	Recommendation for additional medical provision around steep dirt/rocky stages and reassessment of trail design in these technical sections. Improved head injury assessment protocol at EWS races
Roberts, Ouellet et al., 2013 [28]	1995–2009	Southern Alberta, Canada	Retrospective analysis of severely injured cyclists using a trauma database of a Level 1 trauma centre	49	Not specified amateurs	Severe injuries only ISS ≥ 12	87:13 (28)	NR	Severe injuries only, using ISS	Mean transport time to hospital for mountain bikers was 43 min
Romanow, Hagel et al., 2012 [13]	2008–2010	Calgary, Alberta, Canada	Analysis of interviews and questionnaire responses from injured cyclists presenting to 7 emergency depts. near mountain bike parks	409 injured in mountain bike parks	All recreational, excl. racing/competition	All injuries occurring in mountain bike parks	84:16 (19)	NR	Study focus on severe mountain bike park injury, severity based on admission to a hospital inpatient unit	Public health focus: injury prevention strategies should focus on protective equipment worn and limiting speed of riders with trail design as modifiable risk factors
Stoop, Hohenauer et al., 2019 [37]	2017	Switzerland	Retrospective questionnaire of competitors in Swiss Epic mountain bike race	56	Cross-country	All	NR (32.5)	0.39/1000 h exposure time in elite, 0.52/1000 h in amateurs	63%, where severity based on whether concussion/bone injury/joint injury reported	
Taylor and Ranse 2013 [25]	2000–2007	Canberra, Australia	Riders who presented to the first aid station at annual 24 h mountain bike marathon events	596	Cross-country, endurance elites	All	88:12	8.4/1000 bike hours	Severity based on need for transportation to hospital	Noted potential for rural and regional health services to be impacted by competitive events

Terms: NR = not recorded, M:F = male: female, ISS = Injury Severity Score, Level 1 trauma centre provides definitive care from emergency care to rehabilitation.

**Table 4 healthcare-11-03196-t004:** Summary of principal findings and future directions.

Principal Findings from Literature and Risk Factors for Severe Trauma	Potential Future Risk Mitigation Strategies
Inconsistency of reporting on injury severity, lack of description of crash, including use of protective equipment and mechanism of injury	Consistent methods of injury reporting and surveillance by mountain bike parks and health services, including severity scales, large-scale multi-centre studies on mountain biking injuries and competitive event analysis
Limited research on the impact of time to treatment	Research into time to retrieval, treatment, and definitive care to inform medical response and mountain bike park design incorporating emergency access considerations and resourcing of local health services
High social and economic cost of severe injury	Interventions focused specifically on prevention of severe injury
Adverse or unpredictable trail design—high speed into technical downhill features, including trail design in competitive events	Improved trail design to reduce risks associated with severe traumatic injury such as features to help reduce speed into technical sections, comprehensive risk assessment and educating riders on risks
Inadequate trail classification for technical trails	Improved signage, with skill level advised appropriate to trail
Age and poor skill level of rider	Implementation of best practice safe mountain bike riding education for riders and community stakeholders targeted at different age groups and subdisciplines
Subdisciplines involving speed and downhill	Comparison of injury in different mountain biking disciplines, including common mechanisms, injury diagnoses, severity, and outcomes
Bike parks—gravity-focused mountain biking allowing for shuttle access to steeper terrain	Monitor injury rates, implement rider education and training, emergency access preparedness, and health centre resourcing close to mountain bike parks
Electric mountain bike safety	Safety of electric bikes in mountain biking is an area of future research need as this type of riding continues to develop

## Data Availability

The datasets generated and/or analyzed during the current study are available from the corresponding author upon reasonable request.

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
