# Peer review of "Health Service Impacts and Risk Factors for Severe Trauma in Mountain Biking: A Narrative Review"

_healthcare, 2023, doi:10.3390/healthcare11243196_

Round 1

Reviewer 1 Report

Comments and Suggestions for Authors

Dear authors,

This issue is really useful in highlighting the impact that mountain bike injuries, and much more generally accidents, can have on the health system
Congratulations on recognising this and doing something about it.

Nevertheless, there are a few gaps in the essay that should be filled before it can be published.

First of all, the paper is very verbose and challenging to read at places. Please revise it thoroughly to make it more succinct.

I'll break up my recommendations for enhancing the manuscript into sections so that they are more understandable.

Introduction

First of all, I would prefer that you call this part 'introduction' and not 'background'.

Furthermore, given the growing popularity of electric bicycles and scooters, the first section of the introduction ought to address the issue of injuries resulting from traffic accidents, particularly those involving these vehicles in a more organized manner.

The article is incomplete if you only concentrate on injuries caused by mountain bikes in the second place.

Second, the introduction's section on mountain bikes needs to be condensed and analyze the proliferation of these vehicles succinctly; otherwise, it will be very challenging to read.

Methods

If you previously stated that you thought AIS>4 was severe, then I disagree with your statement that ISS>12 should be regarded as serious.

Since many studies in the literature classify ISS 16 as severe, this review can only be compared to other research in this manner.

As a matter of fact, the ISS is determined by taking the square root of the three greatest AIS values.

You would also need to consider ISS>16 to be serious if you thought AIS>4 was severe.

Discussion

The discussion is too long and dense to read; consider rephrasing your points to be clearer and more concise.

4.1 Epidemiology of MTB injuries

The article fails to compare the injuries sustained by mountain bikes with those of other two-wheelers on the road, which is the same issue that I pointed out in the beginning and it still exists throughout the discussion.

In actuality, even traditional bicycles and electric scooters should be taken into consideration in the conversation because they frequently cause minor injuries related to upper limb fractures.

https://doi.org/10.1177/0009922815603676

https://doi.org/10.1007/s12024-022-00546-6

A correlation with other two-wheelers is required even in cases of severe injuries to the head and neck.

https://doi.org/10.1080/15389588.2021.2004311

https://doi.org/10.1007/s12024-022-00477-2

https://doi.org/10.1007/s00701-020-04626-w

Table 1.

It is well done and useful to focus on the types of discipline. However, the sentences need to be more concise or it is difficult to read.

Table 3.

It is well done and useful to focus on the features of injuries. However, the sentences need to be more concise or it is difficult to read.

Comments on the Quality of English Language

It is necessary a moderate editing of English language.

Author Response

Thank you very much for taking the time to review Manuscript ID: healthcare-2719496 Health service impacts and risk factors for severe trauma in Mountain Biking: a narrative review and giving us the opportunity to revise this manuscript. Please find the detailed responses in the attached file and the corresponding revisions/corrections highlighted/in track changes in the re-submitted manuscript.

Reviewer 2 Report

Comments and Suggestions for Authors

Dear Authors,

thank you for the opportunity to review the submitted article entitled: Health service impacts and risk factors for severe trauma in Mountain Biking: a narrative review. Although this type of review may be, in principle, cursory and not free from bias, in the case of the work sent for review, the study appears to be reliable and constructive. The topic of the study may constitute a significant public health problem in the organization of medical care, including the estimation of its costs. The authors' approach to the study, which is practical and characterized by a clinical aspect related to emergency medicine, also seems to be very important.

The article submitted for review, although it does not refer to a health problem with such high incidence and mortality as in the case of other parts of topics usually discussed in scientific literature, seems to be a necessary study.

The methodology of the literature review is at least acceptable. The text of the manuscript itself is organized and written in a clear and transparent way. Apart from minor editorial imperfections (double spaces, lack of spaces before brackets, etc.), the manuscript is of good quality.

There is one question and one strong suggestion for the authors' consideration:

- Do the authors of the literature review know or have they tried to find an answer to the question why, despite the dynamic development of various forms of mountain biking in recent years, as described in the article, studies on severe injuries and related consequences (including those for the health care system and financing) come mainly from sources older than five years? (Table 3: Summary of literature on mountain bike injury rates, types, severity, and health care findings)

- The development of a figure that would refer to the last two subchapters - Conclusions and Future Directions (lines 375-403), could significantly improve the quality of the reader's reception of the article.

Thank you again for the opportunity to review an article on, after all, these very important issues. I sincerely hope that this review will allow you to draw constructive conclusions that will prove useful.

Author Response

(The authors gave the same response as above.)

Reviewer 3 Report

Comments and Suggestions for Authors

The manuscript discusses the increasing participation in mountain biking and the associated risks of severe injury, which can place a burden on health systems. The primary aim of the review is to explore the effects of severe mountain bike injuries on health services and identify risk factors, highlighting gaps in understanding and the need for further research.

In summary, this manuscript sheds light on the growing participation in mountain biking and its associated risks. It underscores the need for more research to better understand and prevent severe injuries, especially in downhill disciplines. Additionally, it emphasizes the importance of consistent reporting of injury severity for a more accurate assessment of the issue.

Highlights:

  1. Relevance of the Topic: Mountain biking is a popular recreational activity, and its increasing participation makes it important to understand the potential health system burdens associated with injuries.
  2. Methodology: The review used a systematic online search of reputable databases, including PubMed and MEDLINE, ensuring a comprehensive exploration of the topic.
  3. Awareness of Health System Impact: The paper highlights the limited documentation of the impacts of mountain biking injuries on health services. It draws attention to the fact that even minor increases in injuries, especially during race events, can strain local health services.
  4. Focus on Downhill Disciplines: The identification of severe injuries being more common in downhill disciplines provides valuable insights into specific areas where injury prevention and healthcare resources could be targeted.

It is noted that there is a spacing issue throughout the manuscript, which need to considered while proof reading. 

Limitations/modifications required :
  1. Methodology: No detailed explanation on search strategy or used key words mentioned. 
  2. Results: Please break them with sub-heading 
  3. Language: It is noted that there is a spacing issue throughout the manuscript, which needs to be considered during proofreading.

Author Response

(The authors gave the same response as above.)

Reviewer 4 Report

Comments and Suggestions for Authors

Dear Respectable Authors

Thank you for considering a significant area of research related to health services in case of trauma for mountain biking. It is a very attractive topic that has been less discussed and due to its growth, it is very important to investigate it. Your review and your results are well-documented. Below are some points that can promote the quality of your article.

- I am a little confused. Your title and aim are contradictory. The effect of which one has been investigated. It seems that the title and aim are opposite. Please clear it.

- The abstract and methods section, please add how you categorized your results or the way you presented your results.

- Please match the title, with the aim stated in the abstract and introduction. 

- If possible, please add your search strategy as a supplementary file. This strategy can be useful for other people who want to do research in this field.

- Please add a column to Table 3 and insert the design of each included study. 

- Please remove the subheadings from your discussion section as there are no such subheadings in the results section. Of course, it is better for the headings of your article to be like this: introduction, main text, and conclusion. In this case, all these sub-headings were placed under the title of the main text.

Cheers

Author Response

(The authors gave the same response as above.)

Round 2

Reviewer 1 Report

Comments and Suggestions for Authors

I thank the authors for responding to my feedback and I also accept the reasons they gave for the points where they decided not to follow the advice I had given. For these reasons, the article is now suitable for publication for me.

Author Response

Thank you for your time in reviewing our revised manuscript and we appreciate your time in helping us to improve our paper.